# Effect of High Hydrostatic Pressure on the Metabolite Profile of Striped Prawn (*Melicertus kerathurus*) during Chilled Storage

**DOI:** 10.3390/foods11223677

**Published:** 2022-11-17

**Authors:** Qiuyu Lan, Silvia Tappi, Giacomo Braschi, Gianfranco Picone, Pietro Rocculi, Luca Laghi

**Affiliations:** 1Department of Agricultural and Food Sciences, Alma Mater Studiorum, University of Bologna, 47521 Cesena, Italy; 2Interdepartmental Centre for Industrial Agrofood Research, Alma Mater Studiorum, University of Bologna, 47521 Cesena, Italy

**Keywords:** fish freshness, high hydrostatic pressure, fish storage, spoilage, metabolomics, ^1^H-NMR

## Abstract

A variety of metabolites contribute to the freshness and taste characteristics of seafood. This study investigated the effects of high hydrostatic pressure (HHP; 400, 500, and 600 MPa) for 10 min) on the metabolome of striped prawn during chilled storage, in relation to microorganisms’ development. All treated samples showed lower viable counts throughout storage compared to the untreated counterparts. The limit of acceptability from a microbiological point of view was extended from 9 to as many as 35 days by 600 MPa treatment. Metabolites were quantified by ^1^H-NMR through a targeted-untargeted metabolomic approach. Molecules linked to nucleotides’ degradation and amines’ anabolism suggested an overall freshness improvement granted by HHP. Notably, putrescine and cadaverine were detected only in untreated prawn samples, suggesting the inactivation of degradative enzymes by HHP. The concentration of molecules that influence umami perception was significantly elevated by HHP, while in untreated samples, the concentration of molecules contributing to a sour taste gradually increased during storage. As metabolomics was applied in its untargeted form, it allowed us to follow the overall set of metabolites related to HHP processing and storage, thus providing novel insights into the freshness and taste quality of striped prawn as affected by high hydrostatic pressure.

## 1. Introduction

The striped prawn (*Melicertus kerathurus*) belongs to a species of prawn autochthonous of the Mediterranean Sea, appreciated and traded for its delicate flavor and pleasant aroma. Striped prawn catches in the Mediterranean Sea recorded more than 6900 tons in 2016, with an additional 300 tons in the Central-Eastern Atlantic [1]. Even if catching practices are evolving toward freezing of the products on the fishing boat, nearly half of the catch is still consumed in its “live, fresh, or chilled” forms [1], prone to the logistic problems connected to high perishability. The primary causes of spoilage, in addition to autolysis, are microorganisms, whose growth negatively impacts shelf life and overall product quality [2].

High hydrostatic pressure (HHP) is a non-thermal technology that has been effectively used on perishable seafoods to reduce the growth of undesirable spoilage microorganisms, thereby substantially extending their shelf life and improving their safety [3]. For these purposes, pressures of 100–600 MPa for 2–10 min at 2–25 °C are generally applied [4,5]. Reyes et al. [6] found that treatment at 450 MPa for 3 min increases the microbiological shelf life of Chilean Jack mackerel from 6 to 29 days, while treatment at 550 MPa for 4 min increases it to 40 d. The extension of the microbiological shelf life is mainly attributed to a reduced initial load of spoilage microflora in the samples. As a consequence, HHP application contributes to delaying the formation of nitrogen-based volatile compounds, mainly trimethylamine (TMA), related to reduced freshness [7].

Microbial growth, and many of its consequences on the sensory characteristics associated by consumers with freshness and high quality, can be conveniently followed by observing the evolution of the fish flesh metabolome, the ensemble of the low-weight molecules it harbors [8]. For example, bacterial growth leads to inosine (Ino) and hypoxanthine (Hx) from adenosine-5′-triphosphate (ATP), adenosine-5′-diphosphate (ADP), adenosine-5′-monophosphate (AMP), and inosine-5′-monophosphate (IMP) [9,10]. The relative amounts of these molecules have been therefore combined for decades in freshness scores, such as the K-value [11]. Moreover, in addition to TMA, the microbial degradation of fish constituents generates biogenic amines and organic acids associated with spoilage [12], as well as unpleasant, bitter-taste-related compounds, including arginine, valine, and methionine. Moreover, some water-soluble, low-weight components are known as taste-active components because they contribute to the specific aspects of seafood’s organoleptic characteristics linked to sweetness, sourness, or umami taste [13].

Unfortunately, all the above-mentioned compounds are characterized by a high variety of functional groups that hinder their rapid and simultaneous determination using a single analytical platform. In this respect, proton nuclear magnetic resonance (^1^H-NMR) can be a convenient tool because the physical principles it relies on may make it intrinsically quantitative regardless of the chemistry of the molecules observed. In fact, numerous works have been devoted to the evaluation of seafood freshness by ^1^H-NMR [8,14,15,16,17].

It is also important to notice that ^1^H-NMR spectra used for analyses targeted toward specific molecules of interest can be simultaneously used for untargeted observations. This is particularly appealing as the consequences of HHP on the fish metabolome are still largely unknown. Therefore, with this work, we sought to perform a combined targeted and untargeted study on the metabolome of prawn flesh treated with HHP during storage, a topic at present limitedly covered by the published literature.

## 2. Materials and Methods

### 2.1. Sample Preparation

The striped prawns (*Melicertus kerathurus*) used in this study were fished in the Adriatic Sea. The company Economia del Mare (Cesenatico, Italy) fast-froze them at −30 °C for 24 h until sample preparation, as typically performed at the industrial level for products intended for raw consumption. For sample preparation, thawing at 4 °C for 16 h was followed by mechanical deboning and removal of the shell. Finally, the striped prawns were manually finely diced, similarly to a fish tartare, into portions (15–20 g) and vacuum-packed in a polypropylene (PP) tray with PP film. A total of 21 packages were created, each containing 6 portions, for subsequent analysis.

### 2.2. HHP Treatment

Vacuum-packaged samples were placed in a 350 L chamber (HPP Italia s.r.l, Parma, Italy) filled with water at 5 °C. The samples were then subjected to 0 (used as a control), 400, 500, and 600 MPa pressure for 10 min. During the rise in pressure, lasting approximately 1 min, adiabatic heating of 3.3 °C every 100 MPa was registered.

### 2.3. Storage

After HHP treatment, all samples were stored at 2 ± 1 °C, and analytical determinations were planned at 1, 6, 9, 14, 21, 28, and 35 days. For each HHP treatment, the effective storage duration was based on the results of microbiological analysis, considering the reaching of a microbial load of 6 log CFU/g as the end of the shelf life. For each treatment, and at each storage time, 2 portions were used for metabolomics analysis by ^1^H-NMR and 3 portions were used for microbiological analysis. Each portion for the same analysis was obtained from different packages.

### 2.4. Microbiological Analysis

By following the rationale of previous works [18], to assess the microbiological quality of packaged striped prawns in relation to the HHP treatment applied, a few selected microbial groups were considered by giving priority to those with the highest expected impact on the metabolome. These were total mesophilic bacteria (TMB; ISO 4833), *Lactobacillus* spp. (ISO 15214), *Pseudomonas* spp. (ISO 13720), *Clostridia* (ISO 7937), total Coliform and *E. coli* (ISO 16649-2), *Staphylococcus aureus* and coagulase positive staphylococci (ISO 6888-1), *Salmonella* spp. (ISO 6579-1:2017/A1:2020), and *Listeria monocytogenes* (ISO 11290-1:2017). The microbial groups were determined according to the corresponding ISO protocols.

### 2.5. Metabolomics Analysis by ^1^H-NMR

A ^1^H-NMR analysis solution was prepared, comprising a 10 mM D_2_O solution of 3-(trimethylsilyl)-propionic-2,2,3,3-d_4_ acid sodium salt (TSP) as a chemical-shift reference (δ −0.017). A 1 M phosphate buffer granted a pH of 7.00 ± 0.02, while 10 μL of NaN3 2 mM avoided microbial proliferation.

By modifying the procedure set up by Ciampa et al. [15], trichloroacetic acid (TCA) extraction was performed by adding 0.5 g of fish muscle to 3 mL of 7% (*w*/*w*) TCA, followed by homogenization by Ultra-Turrax (IKA, Germany) at 14,000 rpm for 20 s. The homogenate was centrifuged at 18,630× *g* for 10 min at 4 °C, and then, 0.7 mL of the supernatant was added with 0.100 mL of a D_2_O solution of 10 mmol/L of 3-(trimethylsilyl)-propionic-2,2,3,3-d4 acid sodium salt (TSP). The pH was adjusted to 7.00 ± 0.02 using 9 mol/L of KOH in an Eppendorf microfuge tube. After centrifuging once more under the above-mentioned conditions, 0.65 mL of the supernatant was transferred to an NMR tube for analysis.

To register ^1^H-NMR spectra, an AVANCE III spectrometer (Bruker, Milan, Italy) was used, operating at a frequency of 600.13 MHz and equipped with Topspin (ver. 3.5) software. ^1^H-NMR spectra were acquired at 298 K using a CPMG pulse sequence, with suppression of the solvent signal, by setting the key parameters as follows: fid constituted by 32 k points for an acquisition time of 2.28 s, 256 scans, 16 dummy scans, 12 ppm of spectral width, and relaxation delay of 5 s. The NMR spectra were pre-processed using Topspin.

Signal assignment to compounds was performed using Chenomx ver 8.3 software (Chenomx Inc., Edmonton, AB, Canada) through comparison with Chenomx (ver. 10) HMDB (release 2) databases. Absolute quantification of molecules was performed in one reference sample by adding 100 µL of maleic acid (9.26 mM) as an internal standard. Spectra from any other sample were adjusted toward the reference by probabilistic quotient normalization (PQN) [19] to compensate for differences in water content. The concentration of each molecule was calculated from the area of one of its signals, measured by global spectra deconvolution, implemented in MestReNova (ver. 14.2.0-26256) software (Mestrelab research S.L. Santiago De Compostela, Spain), by considering a limit of quantification (LOQ) of 5. This was carried out after applying a line broadening of 0.3 and a baseline adjustment by the Whittaker Smoother procedure.

### 2.6. Statistical Analysis

The significance of the differences among groups was tested with SPSS Statistics (ver. 8.0) software by analysis of variance (ANOVA), followed by Tukey’s post hoc test (*p* < 0.05). In R computational language (ver. 4.0.2), Spearman’s correlation tests were performed to determine the relationships between microorganisms and metabolites. For this purpose, *p*-values were adjusted for multiple comparisons according to Benjamini and Hochberg by relying on the R package “*p*.adjust.”

To express the concentration of taste-active molecules in terms of their contribution to sensorial characteristics, their taste-active value (TAV) was calculated as the ratio of the concentration of an individual compound in the striped prawns and its corresponding taste recognition threshold. In this respect, compounds were considered to contribute to the certain taste characteristic of striped prawns when its TAV was higher than 1. Taste thresholds were obtained from the literature [20,21]. The TAVs of individual compounds were then summed up to estimate their overall contribution to the striped prawn taste component (umami, sweet, bitter, sour).

## 3. Results

### 3.1. Characterization of the Striped Prawn Flesh Metabolome

A ^1^H-NMR spectrum representative of the samples observed in the experiment is shown in Figure 1. In total, 53 metabolites were identified in the aqueous extract of striped prawns. The metabolite groups mostly characterizing these spectra (Appendix A) included amino acids, amines, carbohydrates, nucleotides, and organic acids. The most concentrated metabolites were glycine and betaine in each sample. Signals from specific molecules, such as spoilage-related putrescine and cadaverine, were detected only on the last day of storage. The intensities of other signals, such as ethanol, acetate, or ornithine, showed marked differences between treated and untreated samples, especially at the end of the storage period.

To highlight the main trends in the patterns of metabolites, their concentrations were expressed as a distance matrix and then grouped by means of hierarchical clustering. Two heat maps were then generated to show the overall changes in the metabolites due to different processing pressures on day 1 and due to storage time (Figure 2). A visual inspection allowed us to appreciate two main trends. On day 1 (Figure 2a), untreated samples could be easily distinguished from samples treated with any level of pressure. Among samples observed along with storage (Figure 2b), untreated samples collected on days 9 and 14 differed substantially from any other, witnessing a compromised microbial profile.

### 3.2. Effect of HHP on Freshness-Related Metabolites during Storage

#### 3.2.1. Nucleotide Degradation

Among the 53 molecules that we were able to observe by ^1^H-NMR, it was possible to notice the complete set of molecules from which freshness is typically estimated, under the parameter K-value, namely ATP, ADP, AMP, IMP, inosine, and hypoxanthine. As Table 1 shows, there was a gradual increase in the K-value with storage time for all treatments. In detail, the K-value increased for the control group from 39.93% on day 1 to 93.52% on day 9, while it increased from 31.83% on day 1 to 44.13% on day 9 for the samples treated with a pressure of 600 MPa. Intermediate pressures led, as a trend, to intermediate K-values. Focusing on the last day of storage for each treatment, the K-values of the treated samples were all at least 55% lower than the K-values of control samples.

Trends during storage in the nucleotide concentration for each treatment are detailed in Figure 3. Unexpectedly, the total content of ATP, ADP, and AMP in samples treated with a pressure of 400 MPa was systematically lower than in any other and compared to the control group. Moreover, the total concentration of ATP, ADP, and AMP remained low (<1.4 µmol/g) in all samples during the whole storage period. The main differences between control and treated samples was found in IMP, inosine, and hypoxanthine. In control samples, IMP and inosine showed the lowest concentrations, particularly on days 9 and 14, while the hypoxanthine content increased sharply after day 6. Finally, as a general trend, the rate of change in IMP, inosine, and hypoxanthine levels during storage was higher in the control group.

#### 3.2.2. TMAO, TMA, and DMA

Molecules largely quantified for their ability to provide information about fish freshness connected to microbial spoilage are trimethylamine-N-oxide (TMAO) and its volatile catabolites trimethylamine (TMA) and dimethylamine (DMA) [7].

The evolution of TMAO, TMA, and DMA content during storage is shown in Table 2. The TMAO concentration decreased sharply on day 9 in the control group, reaching a concentration of 17.47 mg/100 g of flesh. The same molecule changed moderately with storage in HHP-treated samples, so its levels were all greater than 115 mg/100 g of flesh during storage. A similar but opposite trend was found for TMA. In detail, TMA reached a concentration of 139.1 mg/100 g of flesh in untreated samples after 9 days, while in treated samples, its concentration was under 3 mg/100 g of flesh during the entire storage period. In addition, at each storage time, TMA concentrations in treated samples were always lower than those in the untreated counterparts. DMA was detected in each sample, but no significant differences were highlighted between treated and untreated samples.

#### 3.2.3. Biogenic Amines

Biogenic amines testify to the ongoing proteolysis or amino acid decarboxylase activity of microorganisms during storage and are therefore considered biomarkers of spoilage [12]. The evolution of putrescine, cadaverine, and tyramine content during storage is given in Table 2. Putrescine and cadaverine were detected only in untreated samples on days 9 and 14. The tyramine level could be determined at the end of the storage period in untreated samples and samples treated with 400 MPa HHP, while it was under the limit of detection in samples treated with 500 and 600 MPa HHP.

### 3.3. Effect of HHP on Taste-Active Metabolites during Storage

A part of the metabolites that contribute to the four tastes umami, sweet, bitter, and sour could be detected and quantified by ^1^H-NMR. As shown in Appendix A, on day 1, lactate, IMP, and arginine showed a concentration above the threshold for sour, umami, and bitter tastes, respectively, while glycine, lysine, and alanine exceeded the threshold for sweet taste. Both HHP treatment and storage had an impact on the concentrations of taste-active compounds.

The TAV of lactate systematically decreased below the threshold on the last day of storage in all groups, while IMP and arginine decreased too but never below the threshold. Interestingly, glycine followed a trend like IMP and arginine only in treated samples, while it tended to increase in the control group. The TAV of valine and methionine tended to reach levels above 1 at the end of the storage period for all the treatments tested, while acetate and succinate did so only for untreated samples or for those treated with the lowest pressures.

To get an overview of the effects of these molecules on umami, sweet, bitter, and sour tastes, their sum is shown as radar graphs in Figure 4. The overall umami TAV on day 1, as estimated by metabolomics, was higher in treated than in untreated samples. On the following days, the difference faded away toward values around 6 for each treatment. From the metabolomic perspective, a sour TAV was below the threshold on day 1, while it tended toward values above 1 in untreated samples. Bitterness, as assessed by the combined quantification of nine molecules through ^1^H-NMR, tended to decrease with time as a consequence of each treatment, particularly in untreated or less treated samples.

### 3.4. Effect of HHP Treatment on Other Metabolites

Key metabolites that could be quantified by an untargeted approach are presented in Figure 5. Pyruvate in seafood is mainly produced through pyruvate from glucose, in turn mainly derived from glycogen. Control samples showed a steady decrease, while the opposite was found in samples treated with HHP at 500 and 600 MPa, and an intermediate trend was observed in samples treated with HHP at 400 MPa.

Ethanol decreased from day 1 to day 6 in both treated and untreated samples, and then, it spiked on the last day of storage in the control group and in samples treated with HHP at 400 MPa, reaching, respectively, a concentration 3 and 4 times higher compared to day 1.

Ornithine increased 2-fold on the last day of storage, compared to the corresponding fresh samples, in all treated samples, while it showed a 4-fold increase in the control group.

### 3.5. Effect of HHP on Microorganisms

The effects of HHP treatment on the microbiological quality of packaged striped prawns are reported in Figure 6. In all samples, *Salmonella* spp., *Listeria monocytogenes*, or coagulase-positive staphylococci were never detected during the microbial shelf life of striped prawns. In general, the application of HHP treatment increased the microbiological shelf life of the considered products, and the inactivation effect was more severe at higher pressures.

Regarding untreated striped prawns, initial TMB loads (T1) of control and samples treated at 400 MPa showed values of 4.65 and 2.45 log CFU/g, respectively, while for samples treated at 500 MPa and 600 MPa, mesophilic bacterial counts were below the detection limit (<1 log CFU/g); see Figure 6. Considering a threshold level for acceptability of 6.0 log CFU/g, the TMB in samples treated at 400, 500, and 600 MPa reached such a limit in 21, 28, and 35 days of storage, respectively, compared to 9 days of the control group. A marked difference was evident also in the trends in single species. *E. coli* and *Clostridia* were always under the detection limit for any treatment. In samples treated at 600 MPa, this was true also for total *Coliforms*, *Lactobacilli*, and *Pseudomonas* spp., while intermediate trends could be observed for samples treated at 400 and 500 MPa (Figure 6).

### 3.6. Correlations between Microorganisms and Metabolites

To evaluate the relationship between microorganisms and metabolites, the results of Spearman’s correlation tests between total mesophilic bacteria (TMB) cell loads (log CFU/g) and metabolites are listed in Table 3. TMAO appeared as being negatively correlated with TMB, with a correlation coefficient of −0.66. It is worth noting that TMAO, which produces TMA by bacterial activity, was significantly correlated (*p* < 0.05) with microorganisms. This suggests that the consumption of TMAO is associated with bacteria during the chilled storage of striped prawns. The amino acids could be divided into two groups according to the trend of their correlation. Nine amino acids (tryptophan, phenylalanine, threonine, ornithine, creatine, methionine, isoleucine, valine, and leucine) were positively correlated with TMB, while the opposite could be observed for betaine, glycine, arginine, and sarcosine. All nucleotides (AMP, Ino, IMP) were negatively correlated with TMB, with IMP showing a higher negative correlation of −0.69. This result was consistent with the idea that nucleotides are related to the quality of fish. In addition, significant positive correlations were found between TMB and succinate, acetate, ethanol, and propionate, while opposite correlations were found between TMB and alpha-ketoisovalerate, glycerol, and fumarate.

## 4. Discussion

The effects of HHP treatment on preservation have been observed in a variety of seafood products, with results varying considerably depending on process parameters and also on the seafood species [22,23,24].

Setting up non-thermal methods for the microbiological extension of the shelf life of a fresh food product is a challenging task because it implies a never-perfect compromise between the need for safety and the need to maintain, or even improve, the original organoleptic characteristics. This is especially true when dealing with the application of high hydrostatic pressures to control microorganisms’ proliferation in a product based on fresh striped prawns. Fresh seafood is one of the most prone to rapid microorganisms’ growth. The application of HHP to hinder such growth is still in its infancy, with key pieces of information for optimal application still lacking. This is especially true for the consequences on the food’s molecular profile, as the microorganisms’ metabolism has a direct effect on a wide range of molecules both easily perceived by the consumer and traditionally associated with freshness.

It seems therefore a natural choice to investigate the effects of HHP treatment on seafood by combining microbiological measurements with the quantification of the widest possible set of molecules. In this context, it seems wise to flank observations targeted toward specific molecules, known to have an impact on already established quality indexes, with untargeted metabolomics in order to evidence potential biomarkers or unexpected switches of the microorganisms’ metabolism caused by sub-lethal treatments.

Metabolomics by ^1^H-NMR allowed us to follow the entire set of molecules used to calculate the nucleotide breakdown freshness index K-value, which shows the extent of enzymatic spoilage in hypoxanthine-forming seafood, characterizing those species that contain a low inosine concentration [25,26,27,28]. The differences between untreated and treated samples were striking, with control samples showing 6.7 points of increase per day against 0.35 and 0.43 caused by 500 and 600 MPa treatments, respectively. It is anyway interesting that the 400 MPa treatment caused the K-value to increase by 0.8 points per day. This number is far lower than the one obtained for control samples but anyway double that obtained with stronger treatments, suggesting that the K-value can finely discriminate among treatments of different intensity. Such sensitivity is likely granted by the fact that the K-value combines the concentration of several molecules, so spikes in one are compensated by more stable values of others. A visual impression that the combination of molecules gives finer information than single molecules can be obtained by comparing Table 1 with the trends in single molecules in Figure 3. This figure shows also that the decrease in ATP, ADP, AMP, and IMP does not lead to a corresponding increase in inosine and hypoxanthine, witnessing the evolution of the latter molecules to further degradation products.

The convenience of ^1^H-NMR as an analytical platform for our case is reinforced by the possibility to quantify from the same spectrum used for the K-value also the freshness indicators TMAO and its catabolites, despite the different chemical structure. It is established that microbial activity has the greatest influence on the concentrations of these molecules. Examples can be traced in the work of Prabhakar et al. [29], who found that in Rohu fish stored at 5 °C, TMA passes from values around 0.1 mg/100 g of flesh to approximately 2 mg/100 mg of flesh after 20 days. Our data, anyway, suggest that HHP treatment itself affects, even if at a far lower extent, the concentration of TMAO and TMA. In detail, although control samples show a concentration of TMA of 0.48 mg/100 g of flesh on day 1, treated samples show concentrations from 0.56 to 0.87 mg/100 g of flesh. In parallel, the concentration of TMAO passed from 137.8 mg/100 g of flesh in the control group to values higher than 170 mg/100 g of flesh in treated samples. Changes in the TMA concentration with HHP are well documented in the literature, but they seem to be matrix and temperature related, likely determined by the procedure applied for the extraction from fish flesh. Consistent with our results, Briones-Labarca et al. (2012) found that the application of HHP at 550 MPa for 5 min increases the TMA-N concentration to 0.81 mg/100 g of flesh compared to 0.08 of the control. Contrary to our results, Erkan et al. [22], focusing on horse mackerel, found that applying HHP at 0, 220, 250, and 330 MPa at 7 °C for 5 min leads to TMA-N concentrations of 2.58, 2.45, 1.98, and 1.90, respectively. Intriguingly, Senturk and Alpas [30] found that treating Atlantic mackerel for 5 min with increasing pressures decrease TMA-N at 5 °C but increases it at 15 °C.

Despite the described peculiarities that can be noticed right after HHP treatment, the main differences in TMA and TMAO concentrations can be undoubtedly noticed throughout storage, related to microbial activity. In fact, although in treated samples, TMAO decreased at most by 30% and the TMA concentration did not show appreciable modifications, TMAO decreased and TMA increased by 1 order of magnitude in untreated samples. The complexity of the relationship between treatment and matrix can be observed in this context, too. In fact, although in samples treated at 500 MPa, TMAO decreased by almost 10%, in samples treated at 600 MPa, it showed a decrease close to 30%.

Correlations with total mesophilic bacteria seem to separate amino acids into two groups, with arginine, glycine, betaine, and sarcosine being negatively correlated and nine others showing a positive trend. As many metabolic pathways regulate the concentration of amino acids, it is hard to give a single explanation of the biological reasons. Anyway, Kegg metabolic pathway databases (https://www.genome.jp/kegg/pathway.html, accessed on 31 October 2022) show that sarcosine, betaine, and glycine are doubly linked to glycine through the action of several enzymes, among which are sarcosine dehydrogenase and glycine-N-methyl transferase.

It is intriguing to notice that a single ^1^H spectrum obtained by NMR makes it possible not only to calculate objective indexes of freshness but also to measure molecules with a known direct effect on the consumer’s gustatory perception. The clearest trend that could be observed concerned umami taste on day 1, with pressure increasing the IMP concentration, thus emphasizing umami perception, as observed from a metabolomics point of view. This observation could be rationalized by noticing that high-pressure treatments have been demonstrated to reduce the activity of 5′-nucleotidase (5′-NT) enzyme and nucleoside phosphorylase (NP), which carry out the dephosphorylation of IMP to form Ino and Hx, respectively. Karim et al. [27] reported that high-pressure processing results in significantly lower (*p* < 0.05) mean 5′-NT and NP activity in pressure-treated haddock (*Melanogrammus aeglefinus*) and herring (*Clupea harengus*) compared to the controls. They speculated that high-pressure treatment effectively slows down the conversion of inosine to hypoxanthine.

The possibility to obtain instrumental data on taste preliminary to a proper taste investigation through panelists is compelling. Anyway, it is necessary to emphasize the main limit of our application—that molecules known to have a conspicuous impact on umami taste could not be quantified. This is particularly true for AMP, which could amplify umami taste up to 8-fold [31]. In perspective, the fact that we could actually quantify AMP, but only together with ATP and ADP, allows us to foresee a convenient space for improvements.

Biogenic amines are related to fish spoilage since they accumulate through ongoing proteolysis and amino acid decarboxylase activity of microorganisms in storage [32]. Biogenic amines (putrescine, cadaverine, tyramine, and histamine) have been used as a quality indicator for fish freshness [33,34]. In this study, putrescine, cadaverine, and tyramine were only detected at the end of storage in untreated samples, while putrescine and cadaverine were not detected in treated samples. Our results suggest that the freshness evaluation using biogenic amines as indicators was not relevant for striped prawns because of the low accumulation rate detected. A possible explanation for this might be the low levels of tyrosine, ornithine, and lysine in striped prawns, which degrade to arginine, tyramine, putrescine, and cadaverine, respectively [35]. Another possible explanation for this is the inactivation of microorganisms by HHP, such as *Pseudomonas* spp. and *E. coli*. As reported by previous studies, *Pseudomonas* spp., *E. coli*, *Photobacterium*, *Shewanella* spp., and *Psychrobacter* are responsible for the formation of putrescine and cadaverine in fish and shrimp [35,36,37]. Combined with the observations on microorganisms, this suggests that HHP could reduce the accumulation of biogenic amines and the microbial load. A similar result was reported by Gou et al. in semi-dried squid [38].

The possibility to quantify molecules besides those targeted in connection to previous knowledge, through an untargeted investigation, represents a positive characteristic of ^1^H-NMR. This is particularly evident with ethanol, which showed a marked increase at the end of the storage time in control samples and in samples treated at 400 MPa, thus appearing as a potential convenient biomarker of microbial spoilage. In fact, ethanol is mainly a result of microorganisms’ activity, as documented, among others, by Xu et al. [39]. Another example of potential biomarkers, in line with the observations of Lou et al. [8], is represented by acetate, which increased by 40 and 7 times in samples untreated and treated at 400 MPa but by less than 2 times in the other samples.

## 5. Conclusions

Fish freshness is the result of a complex interaction among a plethora of molecules. Any analytical technique aiming at exploring the topic should, likewise, give the widest possible picture of the fish flesh molecular profile. In this investigation, we observed the consequences of HHP treatments at 400–600 MPa on the metabolome of striped prawn flesh during storage. This was carried out through the calculation of objective indexes of freshness, such as the K-value and the absolute quantification of the amines, key molecules in this respect. From the same ^1^H-NMR spectra, therefore with no extra effort needed, we estimated characteristics linked to perceived freshness, namely umami, sweet, bitter, and sour tastes. More molecules, quantified simultaneously to the so-outlined indexes, were found to be related to total mesophilic bacteria, setting the stage for indexes allowing a direct estimation of microbial spoilage and for a deeper understanding of the underlying mechanisms. Comparisons with literature evidence that the effects of this non-thermal preservation treatment are deeply matrix related, so investigations of this kind should be extended to other kinds of fish and emerging processing technologies.

## Figures and Tables

**Figure 1 foods-11-03677-f001:**
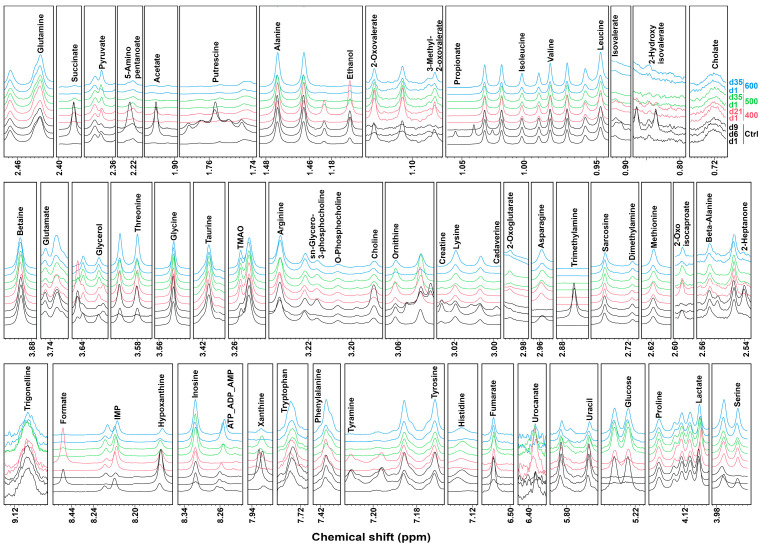
Examples of ^1^H-NMR spectra registered on striped prawn samples analyzed for the investigation.

**Figure 2 foods-11-03677-f002:**
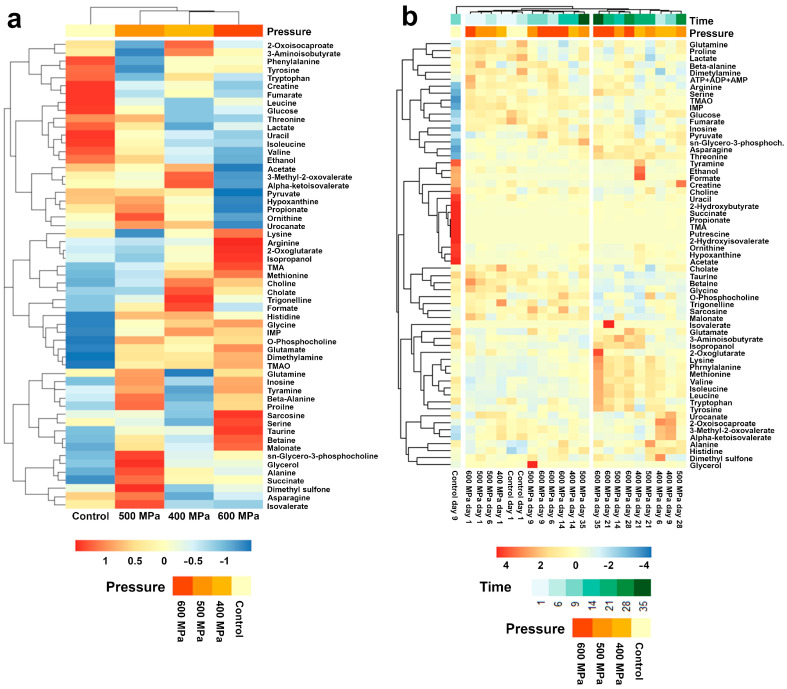
Heatmap visualization of the differential concentrations of metabolites among striped prawn samples subjected to different treatments on day 1 (**a**) and during storage (**b**). Red and green squares highlight metabolites more and less concentrated, respectively, than the average, with the depth of colors indicating greater differences.

**Figure 3 foods-11-03677-f003:**
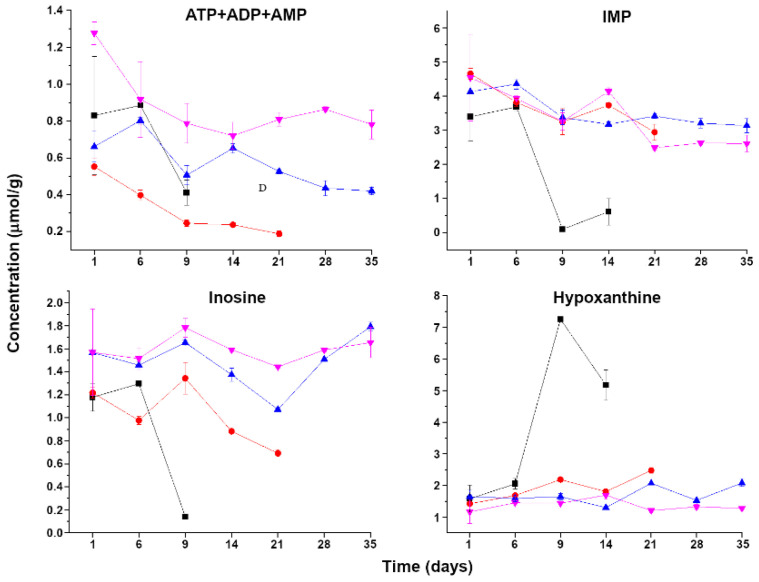
Changes during storage in nucleotide concentrations in striped prawn flesh control samples (black squares) and in samples treated with HHP at 400 (red circles), 500 (blue upward triangles), and 600 MPa (purple downward triangles).

**Figure 4 foods-11-03677-f004:**
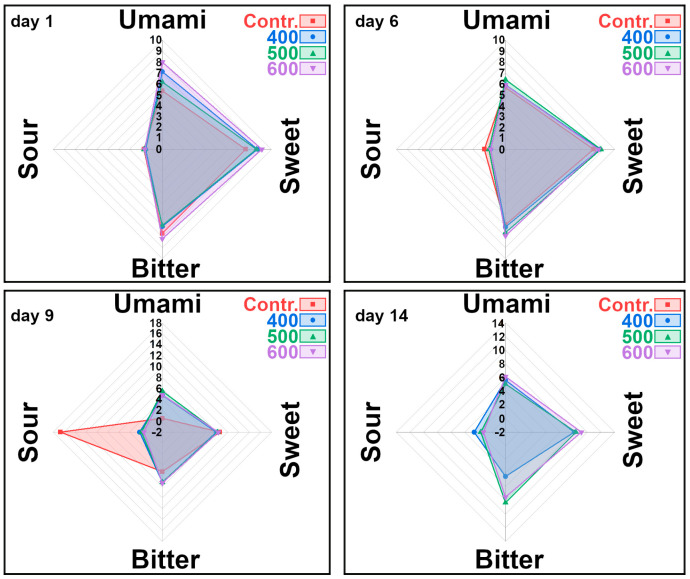
Overall umami, sour, sweet, and bitter taste activity by compounds detected by ^1^H-NMR in control and treated (400, 500, 600 MPa) striped prawns’ samples on days 1, 6, 9, and 14.

**Figure 5 foods-11-03677-f005:**
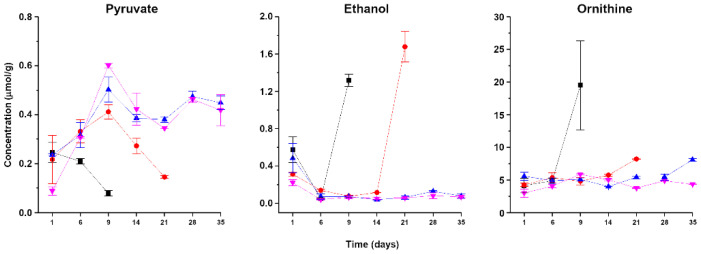
Changes during storage in pyruvate, ethanol, and ornithine concentrations in striped prawn flesh control samples (black squares) and in samples treated with HHP at 400 (red circles), 500 (blue upward triangles), and 600 MPa (purple downward triangles).

**Figure 6 foods-11-03677-f006:**
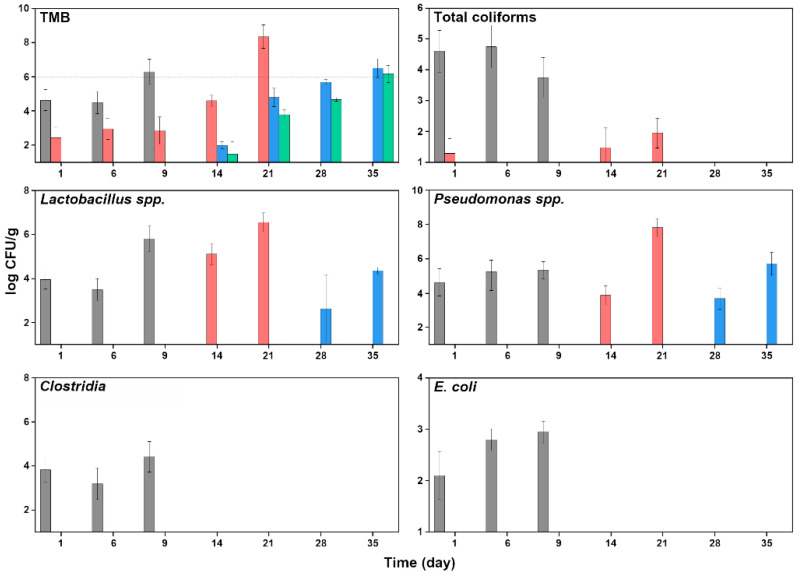
Changes in microbial cell loads (log CFU/g) of total mesophilic bacteria, total Coliforms, *Lactobacillus* spp., *Pseudomonas* spp., *Clostridia*, and *E. coli* during chilled storage of packaged striped prawn untreated (gray) or after treatment with HHP at 400 (red), 500 (blue), and 600 MPa (green).

**Table 1 foods-11-03677-t001:** K-values measured during chilled storage in striped prawns untreated (control) or treated by HHP at 400, 500, and 600 MPa.

Time (Days)	Treatment			
	Control	400 MPa	500 MPa	600 MPa
1	39.93 ± 8.51 ^AB^ *	33.62 ± 1.06 ^AB^	40.11 ± 1.91 ^A^	31.83 ± 0.95 ^B^
6	42.24 ± 0.66 ^A^	38.68 ± 0.29 ^B^	37.13 ± 0.44 ^C^	37.91 ± 1.97 ^AC^
9	93.52 ± 1.23 ^A^	50.28 ± 3.57 ^B^	45.95 ± 2.63 ^BC^	44.13 ± 0.38 ^C^
14		40.36 ± 0.21	41.12 ± 0.03	40.48 ± 1.42
21		50.3 ± 0.83	44.33 ± 0.41	44.78 ± 0.48
28			45.45 ± 1.24	45.56 ± 0.6
35			52.12 ± 1.15	46.32 ± 0.14

* Different capital letters in the same row indicate significant differences (*p* < 0.05).

**Table 2 foods-11-03677-t002:** Amine concentrations (mg/100 g of flesh) during chilled storage of striped prawn untreated (control) or treated with HHP.

Treatments	Time (Days)	TMAO	TMA	DMA	Cadaverine	Putrescine	Tyramine
Control	1	137.8 ± 3.13 ^Ab^ *	0.48 ± 0.04 ^Bc^ **	0.1 ± 0 ^Aa^	- ***	-	-
	6	178.64 ± 0.71 ^Aa^	1.09 ± 0.18 ^Ac^	0.15 ± 0.02 ^Aa^	-	-	-
	9	17.47 ± 2.43 ^Bc^	139.1 ± 0.89 ^Aa^	0.1 ± 0.01 ^Aa^	2.78 ± 0.36	102 ± 13.5	14.19 ± 1.18
400 MPa	1	183.33 ± 14.1 ^Aa^	0.7 ± 0.04 ^ABa^	0.12 ± 0 ^Aa^	-	-	-
	6	141.7 ± 4.1 ^Abc^	0.7 ± 0.05 ^ABa^	0.06 ± 0 ^Bc^	-	-	-
	9	135.93 ± 4.87 ^Abc^	2.55 ± 1.08 ^Ba^	0.1 ± 0 ^Ab^	-	-	-
	14	156.45 ± 5.34 ^Aab^	1.82 ± 0.48 ^Ba^	0.09 ± 0 ^Ab^	-	-	0.69 ± 0.15
	21	125.8 ± 4.61 ^Bc^	0.96 ± 0.01 ^Aa^	0.09 ± 0 ^Ab^	-	-	8.2 ± 0.36
500 MPa	1	173.62 ± 15.48 ^Aa^	0.56 ± 0.05 ^Bb^	0.12 ± 0 ^Aab^	-	-	-
	6	157.63 ± 3.17 ^Aa^	0.52 ± 0.03 ^Bb^	0.12 ± 0 ^ABa^	-	-	-
	9	159.13 ± 2.67 ^Aa^	1.41 ± 0.22 ^Ba^	0.1 ± 0.01 ^Aabc^	-	-	-
	14	150.26 ± 2.72 ^Aa^	0.71 ± 0 ^Bb^	0.1 ± 0 ^Acd^	-	-	-
	21	148.7 ± 4.65 ^Aa^	0.79 ± 0.03 ^Ab^	0.1 ± 0 ^Abc^	-	-	-
	28	138.74 ± 8.26 ^a^	0.4 ± 0.01 ^b^	0.08 ± 0 ^d^	-	-	-
	35	156.72 ± 0.35 ^a^	0.47 ± 0.03 ^b^	0.11 ± 0 ^ab^	-	-	-
600 MPa	1	186.41 ± 18.67 ^Aa^	0.87 ± 0.01 ^Aa^	0.12 ± 0.02 ^Aa^	-	-	-
	6	161.97 ± 15.03 ^Aabc^	0.5 ± 0.02 ^Bd^	0.13 ± 0.02 ^ABa^	-	-	-
	9	151.44 ± 6.29 ^Aabc^	0.8 ± 0.03 ^Bab^	0.11 ± 0 ^Aa^	-	-	-
	14	173.41 ± 4.35 ^Aab^	0.69 ± 0.02 ^Bbc^	0.1 ± 0 ^Aa^	-	-	-
	21	115.76 ± 6.1 ^Bc^	0.59 ± 0.03 ^Bcd^	0.1 ± 0.01 ^Aa^	-	-	-
	28	120.67 ± 9.25 ^c^	0.5 ± 0.05 ^d^	0.11 ± 0 ^a^	-	-	-
	35	130.21 ± 3.59 ^bc^	0.64 ± 0.02 ^d^	0.1 ± 0 ^a^	-	-	-

* Different capital letters (A, B, C) in the same column indicate significant differences on the same day among treatments (*p* < 0.05). ** Different lowercase letters (a, b, c) in the same column indicate significant differences among different storage times for the same treatment (*p* < 0.05). ***—below the LOQ.

**Table 3 foods-11-03677-t003:** Correlations between total mesophilic bacteria (TMB) and the metabolite concentrations in striped prawns during chilled storage.

	Metabolite	Correlation	Adjusted *p*-Value
Amines			
	Trigonelline	−0.41	1.46 × 10^−2^
	TMAO	−0.66	8.72 × 10^−6^
Amino Acids			
	Arginine	−0.59	1.21 × 10^−4^
	Glycine	−0.51	1.29 × 10^−3^
	Betaine	−0.47	4.77 × 10^−3^
	Sarcosine	−0.45	5.64 × 10^−3^
	Ornithine	0.34	4.87 × 10^−2^
	Creatine	0.36	3.33 × 10^−2^
	Tryptophan	0.38	2.38 × 10^−2^
	Threonine	0.43	8.70 × 10^−3^
	Methionine	0.46	5.64 × 10^−3^
	Phenylalanine	0.52	1.17 × 10^−3^
	Leucine	0.59	1.21 × 10^−4^
	Isoleucine	0.63	3.49 × 10^−5^
	Valine	0.73	4.87 × 10^−7^
Organic acids			
	Fumarate	−0.51	1.29 × 10^−3^
	Succinate	0.59	1.21 × 10^−4^
	Acetate	0.56	3.38 × 10^−4^
Nucleotides			
	AMP	−0.5	2.01 × 10^−3^
	Inosine	−0.45	5.64 × 10^−3^
	IMP	−0.69	2.40 × 10^−6^
Other			
	Glycerol	−0.4	1.74 × 10^−2^
	Alpha-ketoisovaleric acid	−0.4	1.59 × 10^−2^
	Ethanol	0.34	4.87 × 10^−2^
	Propionate	0.44	8.12 × 10^−3^

## Data Availability

Data in contained within the article.

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
