# Peer review of "Effect of High Hydrostatic Pressure on the Metabolite Profile of Striped Prawn (Melicertus kerathurus) during Chilled Storage"

_foods, 2022, doi:10.3390/foods11223677_

Round 1

Reviewer 1 Report

The manuscript received for review investigates the effect of high hydrostatic pressure on metabolic profiling of striped prawn dring chilled storage.

Manuscript is clearly and adequately written.

The title of the manuscript is informative provides sufficient data, although it needs some changing.

Introduction section comprehensive, with enough contemporary literature. The goal of the research needs corrections

The Materials and Methods section describes samples’ preparation process and all conducted analysis in sufficient details. Some needed elaborations are needed.

The results and discussion section should be merged in one section. Results are comprehensive, while discussion is adequate with numerous referencing to other authors’ findings.

Conclusion section could be expanded with more detail regarding amount of data presented in results and discussion section.

Reviewer recommendation: Major revision.

Detail comments are noted in pdf file.

Author Response

We thank the reviewer for the encouraging comments. As the reviewer wrote the comments in the pdf file, we report them here, in the attachment, to address them one by one

Reviewer 2 Report

This manuscript reported the effect of high hydrostatic pressure (HHP) treatment on the prawn during cold storage on molecular and microbiological basis. The results shown in this manuscript are clearly presented and also useful and informative for the field.

Specific comments:
1. The authors should specify the temperature during HHP treatment, whether it was controlled or not, because the temperature increased during HHP treatment, especially 500 MPa or 600 MPa treatments.

2. The authors mentioned that ISO protocols were used for detecting Listeria monocytogenes and Salmonella spp. How about the other bacteria? If the authors did not use the official and/or standard protocols for other bacteria, please specify the reasons.

3. Why didn’t the authors include Vibrio spp. for microbiological testing? Vibrio spp. were one of the most important pathogens for the seafood.

4. The authors mentioned that ethanol was mainly a result of microorganisms’ activity in the Discussion, however, Xu et al. [38] mentioned that “both microbial and endogenous lipases contributed to the FFA liberation in fermented fish while endogenous lipases play a major role.”

Author Response

(The authors gave the same response as above.)

Reviewer 3 Report

Article

Tittle: Effect of high hydrostatic pressure (HHP) on metabolic profiling of striped prawn (Melicertus kerathurus) during chilled storage

Journal: Foods

Here are some comments to improve the quality of the paper.

·         Page 1, in tittle the word “metabolic profiling” doesn’t go in relevance with the study as it depicts the reactions associated with metabolism whereas the study is about metabolites. Therefore, “metabolite profile” will be more suitable in tittle.

·         13. high hydrostatic pressure (400, 500 and 600 MPa.  

·         line no. 34 and 36, the reference (1) is cited twice in continuity. Cite it once.

·         42. Please check these paper and update your data.

Emerging trends for nonthermal decontamination of raw and processed meat: Ozonation, high-hydrostatic pressure and cold plasma.

Effect of high–hydrostatic pressure processing and sous-vide cooking on physicochemical traits of Biceps femoris veal patties

High-pressure processing of fish and shellfish products: safety, quality and research prospects.

·         line no. 44, correctly cite the reference. It should be “Reyes et al. (2015)”.

·         line no. 75, clarify the aim/objectives of the study and briefly describe it. The description is not giving a clear insight of the study aim and causing confusion.

·         line no. 80, the freezing and thawing of the product cease several microbial, metabolites and enzymatic activity that affect the shelf-life and storage. Does this fact interferes the study results of HHP effects during chilled storage?

·         line no. 83, why you specifically selected the polypropylene tray with PP films despite its health compromising drawbacks?

·         line no. 99-101, is there any criteria for the selection of these few bacterial species monitoring in the study?

·         line no. 124, mention the year with citation.

·         line no. 172, there is a difference in the molecules between treated and untreated samples. How? Justify the mechanism.

·         line no. 204, in table 1 why the K-value is fluctuating as value increase up to day 9 and decrease on day 14 and then again start to increase onwards? Justify it.

·         line no. 237, what do you mean by “good indicator of spoilage”?

·         line no. 252, use past tense instead of “can be”.

·         line no. 272, what was the bitterness tending in fully treated samples? If increasing then isn’t it a bad trend?

·         line no. 319-323, amino acids are grouped into two on the basis of their correlation with TMB. Justify the difference among correlations.

·         Does the HHP treatment affect the texture of the samples during the study?

·         Page 13, elaborate conclusion more briefly and add up more concluding remarks.

·         Extensive grammatical and English language mistakes throughout the manuscript. Improve scientific writing.

Author Response

We thank the reviewer for the encouraging comments. As the reviewer wrote the comments in the pdf file, we report them here, in the attachment, to address them one by one.

Round 2

Reviewer 1 Report

The quality of the research is improved. The manuscript is suitible for publication.